# Spin-momentum locked interaction between guided photons and surface electrons in topological insulators

Siyuan Luo[1,2], Li He[1,3] & Mo Li [1]

The propagation of electrons and photons can respectively have the spin-momentum locking effect which correlates spin with linear momentum. For surface electrons in three-dimensional topological insulators (TIs), their spin is locked to the transport direction. Analogously, photons in optical waveguides carry transverse spin angular momentum which is also locked to the propagation direction. A direct connection between electron and photon spins occurs in TIs due to spin-dependent selection rules of optical transitions. Here we demonstrate an optoelectronic device that integrates a TI with a photonic waveguide. Interaction between photons in the waveguide and surface electrons in a $Bi_2Se_3$ layer generates a directional, spin-polarized photocurrent. Because of spin-momentum locking, changing light propagation direction reverses photon spin and thus the direction of the photocurrent. Our device represents a way of implementing coupled spin–orbit interaction between electrons and photons and may lead to applications in opto-spintronics and quantum information processing.

[1] Department of Electrical and Computer Engineering, University of Minnesota, Minneapolis, MN 55455, USA. [2] Institute of Fundamental and Frontier Sciences, State Key Laboratory of Electronics Thin Films and Integrated Devices, University of Electronics Science and Technology of China, Chengdu 610054, China. [3] School of Physics and Astronomy, University of Minnesota, Minneapolis, MN 55455, USA. Siyuan Luo and Li He contributed equally to this work. Correspondence and requests for materials should be addressed to M.L. (email: moli@umn.edu)

The electric field of a non-paraxial optical beam or laterally confined optical modes in fibers or waveguides is no longer purely transverse but has a longitudinal field component, which has a $\pm\pi/2$ phase shift relative to the transverse field component. Therefore, the total electric field is spinning along an axis transverse to the light propagation direction and thus elliptically polarized in the propagation plane[1,2]. Consequently, the corresponding photons carry transverse spin angular momentum (SAM)[3]. The handedness of the polarization and the transverse SAM are locked to the propagation direction, meaning that their signs reverse with the direction of propagation, as schematically illustrated in Fig. 1a for a waveguide mode[4–7]. This effect is analogous to the quantum spin Hall effect and the spin-momentum locking effect occurring for the surface electrons in topological insulators[8–12], as depicted in Fig. 1b, albeit the optical systems (waveguides, fibers) are topologically trivial.

It is intriguing to consider that whether such two conceptually related phenomena for electrons and photons could be coupled through a new way of light–matter interaction. Such an interaction may enable the conversion between the photon momentum to the electron spin and vice versa. Indeed, the strong spin–orbit coupling in topological insulator (TI) materials such as $Bi_2Se_3$ leads to lifted spin degeneracy and consequently selection rules for interband optical transitions that are dependent of the electron spin. As a result, circularly polarized light can selectively excite surface electrons with one type of spin and generate a directional, spin-polarized surface current[13–18]. This spin-polarized and helicity-dependent photoexcitation is illustrated in Fig. 1c. Because the photocurrent is generated without a bias voltage, it is named circular photogalvanic effect (CPGE), which has also been observed in other material systems with spin–orbit coupling[19–22]. However, only in TIs the effect is due to topologically protected surface states.

Consider the so-called quasi-transverse electric (TE) and transverse magnetic (TM) modes of a waveguide made of silicon, as simulated in Fig. 1d, which is 1500 nm wide and 220 nm high and cladded on all sides with silicon dioxide. The modes have a considerable longitudinal field component with a $\pm\pi/2$ phase difference to the transverse field component, and thus are elliptically polarized in the propagation plane with transverse SAM. The sign of the phase difference, which determines the handedness of the elliptical polarization and the sign of the transverse SAM, depends on the propagation direction—the optical spin-momentum locking effect[4,7]. In Fig. 1d, we plot the calculated transverse component (the $x$-component, Fig. 1a) of the electric SAM density[23], defined as $\mathbf{S} = -[i\varepsilon_0/2\omega]\mathbf{E}^* \times \mathbf{E}$, where $\varepsilon_0$ is vacuum permittivity and $\omega$ is the optical frequency of the TE and TM modes. It is apparent that, on the top of the waveguide, the transverse electric SAM density of the TM mode is more significant, more than 100 times higher than that of the TE mode. Now consider a layer of TI material placed on the top of the waveguide, as marked with the dashed line in Fig. 1d. Because the photons in the TM mode are elliptically polarized, through the CPGE effect they will induce a photocurrent flowing in the longitudinal direction (the $z$-axis) with spin polarization along the transverse direction (the $x$-axis). In this way, the CPGE interaction couples the spin-momentum locking of the TM mode photons with the surface electrons in the TI. Reversing the propagation direction of the light in the waveguide will reverse the handedness of the elliptical polarization, the CPGE photocurrent direction and the spin polarization.

In this work, we integrate a layer of TI $Bi_2Se_3$ directly on a photonic waveguide to realize a helicity-dependent interface between the topological surface electrons and the guided photons. We show that reversing the propagation direction of the light in the waveguide changes the sign of the circular photogalvanic current generated in the TI, which should be highly spin polarized. Therefore, such a TI-waveguide integration makes an opto-spintronic device, the first of its kind, that converts the propagation path information (forward or backward) of photons

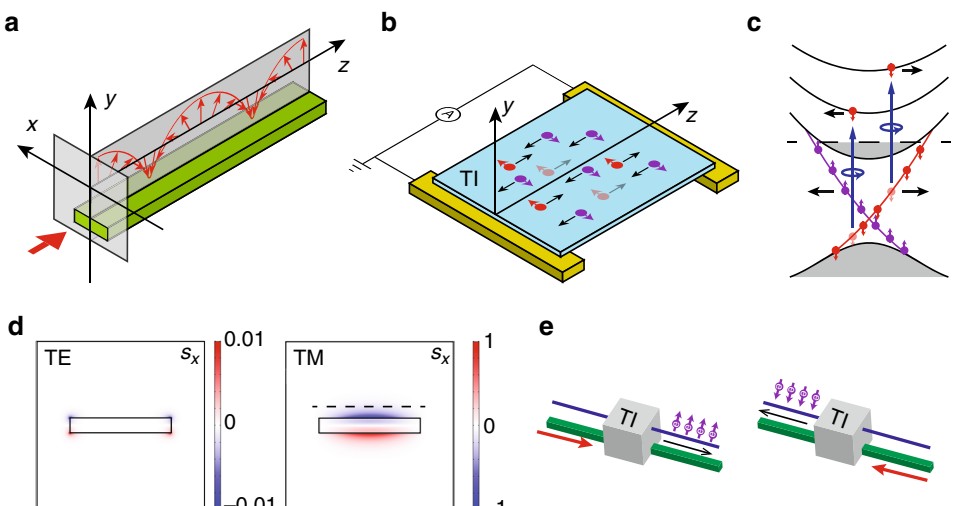

**Fig. 1** Spin-momentum locking for photons and electrons. **a** In a waveguide mode, the evanescent electric field is spinning in the propagation plane (the $y$–$z$ plane) and therefore the guided photons have a SAM polarized along the transverse direction (along the $x$-axis). The direction of the SAM is locked to the propagation direction: along $+x$ for propagation long $+z$ and reverse to $-x$ for propagation long $-z$. **b** The spin of the surface electrons in a topological insulator is locked to the transport direction. **c** The selection rules of optical transitions in TI with lifted spin degeneracy lead to spin-dependent photoexcitation. Circularly polarized light will selectively excite electrons with one type of spin from the surface states to the bulk bands, generating a spin-polarized photocurrent flowing in a direction determined by the spin polarization. This is the CPGE effect. **d** The $x$-component of SAM density ($S_x$) for the TE and TM mode of a silicon waveguide (1500 nm wide and 220 nm high). On the top of the waveguide, $S_x$ of the TM mode is more than 100 times higher than that of the TE mode. Therefore, the TM mode has a more significant helicity-dependent effect in its evanescent field. **e** The proposed device integrates the TI on a waveguide to explore the spin-momentum locking of electrons and photons. It outputs a spin-polarized current with its spin polarization and direction determined by the light input direction

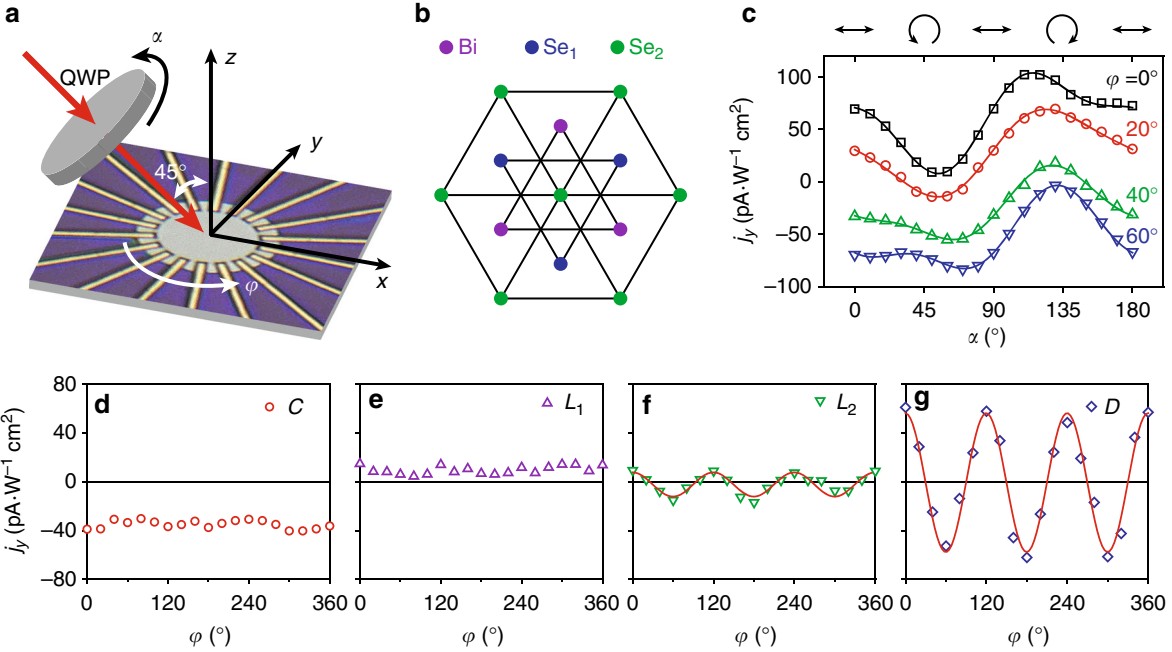

**Fig. 2** Anisotropy of the photogalvanic effects in $Bi_2Se_3$. **a** Measurement scheme. The polarization of the incident light is changed by rotating a quarter waveplate with angle $\alpha$. The $Bi_2Se_3$ is patterned to a circular disk. Nine pairs of electrodes are used to measure photocurrent generate along different directions when the sample is rotated with angle $\varphi$. **b** The lattice structure of the $Bi_2Se_3$ crystal showing threefold rotational symmetry along the $z$-axis. **c** Polarization-dependent photocurrent collected with electrode pair positioned along the $y$-axis when the sample is rotated to representative values of $\varphi$. **d**–**g** Parameters ($C$, $L_1$, $L_2$, $D$) extracted from the measurement in (**c**) as a function of $\varphi$. While $L_2$ and $D$ show clear anisotropy with threefold rotational symmetry in agreement with the crystal structure of $Bi_2Se_3$, $C$ and $L_1$ show no obvious angular dependence. The fitting errors for all the coefficients are very small and hence not shown here

to the spin polarization of electrons in a directional photocurrent, as schematically illustrated in Fig. 1e. Note that because when both electron and photon spins are considered, the device under forward light input situation can be transformed to the situation of backward light input with a $\pi$ rotation, such a device is achiral.

## Results

**Anisotropic photogalvanic effects**. To experimentally demonstrate such a device, we use films of topological insulator $Bi_2Se_3$ exfoliated from a bulk crystal. We first investigate the photogalvanic effects in the samples using a free-space optics configuration similar to the previous studies[14,16,17], but with the difference that infrared light (1.55 μm in wavelength) is used instead of visible light. The polarization of the incident light is varied from linear to circular by rotating a quarter waveplate with angle $\alpha$, while the incident angle is fixed at 45° (in $x$–$z$ plane) to the sample surface. The measurement configuration is shown in Fig. 2a. Noting that, in the presence of a surface, $Bi_2Se_3$ thin film has a $C_{3v}$ crystalline structure which has a threefold rotation symmetry around the $z$-axis (Fig. 2b), and we investigate the anisotropy of its photogalvanic effects. Since the photogalvanic current consists of contributions from both the surface and the bulk, the study of the anisotropy of different contributions can help reveal their mechanisms. To this end, we patterned the $Bi_2Se_3$ flake into a circular shape using ion milling and deposited 18 electrodes positioned in a rotation around the sample. The photocurrent was collected from the pair of electrodes positioned along the $y$-axis when the sample was rotated in directions specified with the azimuthal angle $\varphi$, as shown in Fig. 2a. To minimize the thermoelectric effect due to non-uniform heating of the sample, we focused the laser to a spot of 330 μm in diameter, much larger than the size of the $Bi_2Se_3$ sample which has a diameter of 10 μm. The laser power level is fixed at 45 mW, at which

the photoresponse of the device is in the linear regime (Supplementary Note 1 and Fig. 2). The results of polarization-dependent photocurrent with $\varphi = 0°$, 20°, 40° and 60° are plotted in Fig. 2c, which show similar variation with polarization as reported in the previous works[14,16,17]. The polarization dependence also exhibits clear difference when the photocurrent is collected with different sample rotation angle $\varphi$, manifesting the anisotropy. We fit the results with the model that includes four components of distinctive polarization dependence:[14]

$$j_y(\varphi, \alpha) = C(\varphi)\sin 2\alpha + L_1(\varphi)\sin 4\alpha + L_2(\varphi)\cos 4\alpha + D(\varphi)$$

(1)

The coefficient $C$ corresponds to the CPGE, coefficient $L_1$ and $L_2$ correspond to linear photogalvanic effect (LPGE) but possibly with different origins and $D$ is the polarization-independent photocurrent that includes a thermoelectric contribution. We fit the measurement results obtained at each $\varphi$ and extract the parameters ($C$, $L_1$, $L_2$, $D$). The angular-dependent results are plotted in Fig. 2d–g. The results for $L_2$ and $D$ clearly exhibit a threefold rotation symmetry with a period of 120°, in agreement with the symmetry of the $Bi_2Se_3$ crystal. This agreement corroborates with the conclusion of previous temperature-dependent studies that $L_2$ and $D$ stem from a similar origin of the bulk crystal[14]. In clear contrast, the results of $C$ and $L_1$ show indiscernible angular dependence. Because the helical Dirac cone for the surface states appears at the $\Gamma$ point in $k$-space, at the low energy limit, rotational symmetry is expected for interband transitions that involve the surface states. Therefore, $C$ and $L_1$ are likely to share a common origin of interband transitions from the occupied surface states to the bulk bands above (Fig. 1c), given that the exfoliated $Bi_2Se_3$ sample is $n$-doped and the photon energy of 0.8 eV is much larger than the bulk bandgap of 0.3 eV.

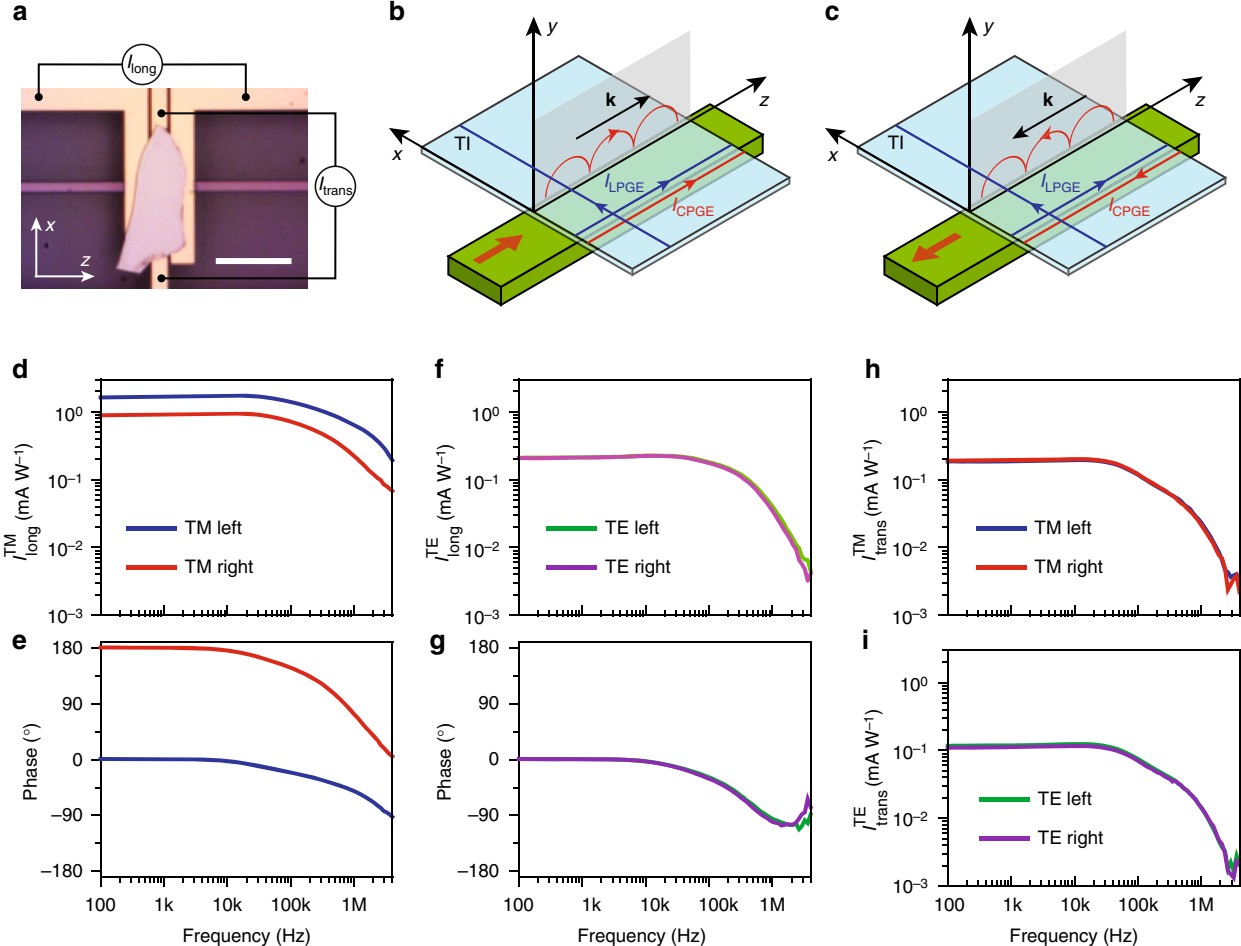

**Fig. 3** Directional photogalvanic interaction between the waveguide mode and the surface electrons in the TI. **a** Optical microscope image of the TI waveguide-integrated device and the measurement scheme. Two pairs of electrodes are fabricated to collect photocurrent in the longitudinal direction (along z-axis) and the transverse direction (along x-axis). The scale bar is 10 μm. **b, c** The CPGE photocurrent $I_{CPGE}$ generated in the TI by the TM mode of the waveguide is only along the longitudinal direction. Because of the photonic spin-momentum locking, $I_{CPGE}$ changes direction when the light propagation direction in the waveguide is changed from forward (**b**) to backward (**c**). In contrast, the LPGE photocurrent has both longitudinal and transverse components which are independent on the light propagation direction. **d, e** Frequency response of the amplitude (**d**) and the phase (**e**) of the longitudinal photocurrent $\left(I_{long}^{TM}\right)$ when TM mode is launched in the waveguide. When the light propagation direction is reversed from leftward to rightward, the sign of the photocurrent reverses, as seen from the 180° change in the phase response, and the amplitude of the photocurrent also changes. **f–i** The longitudinal photocurrent of the TE mode $\left(I_{long}^{TE}\right)$ and (amplitude in e, phase in g), the transverse photocurrent of the TM mode $\left(I_{trans}^{TM}\right)$ (**h**) and the TE mode $\left(I_{trans}^{TE}\right)$ (**i**) all show negligible dependence on the light propagation direction

In addition, photon drag effect may also cause helicity-dependent photogalvanic current; however, recent measurement by varying the incident angle has ruled it out[24]. Our angular-dependent photogalvanic current measurement is useful to clarifying the origins of different contributions and shows that the overall photogalvanic current varies with the crystalline orientation due to the anisotropic contribution from the bulk.

**Waveguide-integrated Bi₂Se₃ device.** We next integrate an exfoliated $Bi_2Se_3$ flake on a silicon waveguide to demonstrate the device described in Fig. 1e. Figure 3a shows the optical microscope image of a representative device (Device C). The $Bi_2Se_3$ flake is transferred onto the top of the waveguide which is cladded with a 150 nm thick layer of silicon dioxide. This cladding layer is intentionally designed to be thick to avoid significant disturbance of the waveguide mode by both the TI and the electrical contacts, but also allows the evanescent field of the mode to interact with the TI. Two pairs of electrodes underneath the flake are used to collect the photocurrent generated from the bottom surface and the bulk of the TI. One electrode pair is positioned (with a

separation of $l = 3$ μm for Device C) along the longitudinal direction (z-axis) of the waveguide and the other pair along the transverse direction (x-axis). We design the integrated photonic circuit (Supplementary Fig. 1) to allow coupling of either TM or TE modes to the waveguide, and symmetrically from either the left or the right ends. Therefore, overall there are eight different measurement configurations for the TM or the TE modes propagating either leftward or rightward along the waveguide with either the longitudinal ($I_{long}$) or the transverse current ($I_{trans}$) being collected.

Figure 3b, c illustrates the situations when the TM mode is launched into the waveguide. As shown in Fig. 1d, the photons in the TM mode are elliptically polarized and carry transverse electric SAM (along x-axis), and through the CPGE effect they will selectively excite surface electrons with their spin aligned with the optical SAM and induce a spin-polarized net current $I_{CPGE}$ flowing in the longitudinal direction (z-axis) due to the spin-momentum locking effect of topological surface states. Here, the CPGE effect is due to the SAM of the waveguide TM mode in the transverse direction, rather than the longitudinal SAM as in

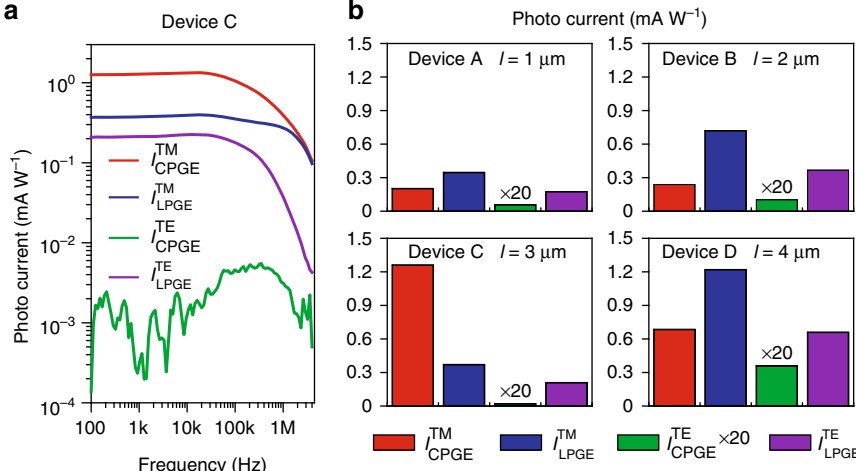

**Fig. 4** Contributions of circular and linear photogalvanic currents. **a** Frequency response of the extracted CPGE and LPGE contributions of photogalvanic current for TM and TE modes in Device C. **b** CPGE and LPGE contributions for TM and TE modes in four devices with different TI lengths $l$ between the electrodes from 1 to 4 μm. A large variation of the ratio between the CPGE and the LPGE contributions is attributed to the high anisotropy of the LPGE, which is sensitivity to the crystalline orientation of the $Bi_2Se_3$ flake to the waveguide

the free-space laser beam (Fig. 2). Importantly, when the propagation direction of the TM mode is reversed (equivalent to apply time-reversal operation and change wave vector **k** to −**k**), the polarity of **S** also reverses—the optical spin-momentum locking—and the resulting $I_{CPGE}$ will flow in the opposite direction, as illustrated in Fig. 3c. Such a longitudinal photocurrent with its direction dependent on the light propagation direction is thus the hallmark of the spin-momentum locked interaction between the guided optical mode and the topological surface states. Other than CPGE, other effects including LPGE and thermoelectric effect can also generate photocurrent so the total longitudinal photocurrent should be expressed as: $I_{long} = I_{CPGE} + I_{LPGE} + I_{th}$. The current produced by LPGE can have both transverse ($x$-) and longitudinal ($z$-) components but should be independent of the light propagation direction (Fig. 3b, c). The thermoelectric effect due to the temperature gradient along the waveguide may induce a longitudinal photocurrent ($I_{th}$) that also reverses sign with the light propagation. However, this thermal effect should work in the same way for both TM and TE modes; the latter, however, should produce negligible longitudinal $I_{CPGE}$ because of its negligible $S_x$ in its evanescent field (see Fig. 1d). Therefore, through analyzing the results obtained with the different measurement configurations in our device, contributions from different mechanisms can be unambiguously distinguished from each other.

Figure 3d–i shows the photocurrent, measured with different experiment configurations, as a function of frequency when the laser is modulated with an electro-optic modulator and coupled into the waveguide through integrated grating couplers for TE and TM modes. A lock-in amplifier is used to measure the photocurrent and its phase response, which is important in determining the direction of the photocurrent. The most notable results are shown in Fig. 3d, e when the TM mode is launched into the waveguide from either the left or the right side. The longitudinal current $I_{long}^{TM}$ and its phase show a strong dependence on the light propagation direction. Specifically, when the light propagation direction is changed from leftward to rightward, the sign of $I_{long}^{TM}$ reverses, as can be seen from the 180° change in the phase response, along with a change in the magnitude. In contrast, in Fig. 3f, g, when TE mode is launched in the waveguide, reversing the light propagation direction has a negligible effect on both the magnitude and the phase of $I_{long}^{TE}$.

$I_{long}^{TE}$ is expected to include contributions from both LPGE effect $\left(I_{LPGE}^{TE}\right)$ and thermoelectric effect ($I_{th}$) but negligible contribution from the CPGE effect $\left(I_{CPGE}^{TE}\right)$ (Fig. 1d). Considering the thermoelectric effect, the temperature gradient is along the $z$-axis in the sample and the induced $I_{th}$ should be reversed when the light propagation direction is reversed. However, in Fig. 3f, g, $I_{long}^{TE}$ shows unnoticeable difference between the opposite light propagation directions. Therefore, we can conclude that the thermoelectric effect is insignificant in our device, possibly because of the small device size and efficient heat sinking. After ruling out the thermoelectric contribution, for the TM mode, we can express the longitudinal photocurrent as $I_{long}^{TM\pm} = \pm I_{CPGE}^{TM} + I_{LPGE}^{TM}$, where $I_{CPGE}^{TM}$ is the CPGE photocurrent which flows along the light propagation direction (marked with ±), and $I_{LPGE}^{TM}$ is the LPGE photocurrent which is independent of the light propagation direction. Similarly, for the TE mode, $I_{long}^{TE\pm} = \pm I_{CPGE}^{TE} + I_{LPGE}^{TE}$. The transverse photocurrents along the $x$-axis generated by the TM $\left(I_{trans}^{TM}\right)$ and the TE modes $\left(I_{trans}^{TE}\right)$, as shown in Fig. 3h, i, provide good control results. Because both the TM and TE modes are linearly polarized in the $x$–$y$ plane, there is zero $z$-component of SAM ($S_z$) and hence no CPGE photocurrent along the transverse $x$-direction. The symmetry of the device also minimizes the thermoelectric contribution. In addition, thermoelectric current also has a much lower roll-off frequency than photogalvanic current (Supplementary Note 2 and Fig. 3) which cannot be observed in Fig. 3. Therefore, we can conclude that $I_{trans}^{TM}$ and $I_{trans}^{TE}$ consist of only LPGE photocurrent (Fig. 3b, c), which does not change with the light propagation direction. This is clearly shown in Fig. 3h, i. The difference between the magnitude of $I_{trans}^{TM}$ and $I_{trans}^{TE}$ is attributed to the different field intensities of the two modes at the TI layer. Above results and analysis show that, among all of the configurations, only the longitudinal photocurrent generated by the TM mode $\left(I_{long}^{TM}\right)$ uniquely shows a strong dependence on the light propagation direction due to the spin-momentum locked interaction between the photon transverse spin and the TI surface electrons.

## Discussion
We can further extract and quantify the contributions of the CPGE and LPGE effects in above results by using the reciprocal

symmetry of the device such that: $I_{CPGE}^{TM} = \left(I_{long}^{TM+} - I_{long}^{TM-}\right)/2$ and $I_{LPGE}^{TM} = \left(I_{long}^{TM+} + I_{long}^{TM-}\right)/2$. The results obtained from Device C are summarized in Fig. 4a and show that, for the TM mode, the CPGE contribution is more than 3 times larger than the LPGE contribution. The dominance of the CPGE effect results in the reversal of the total photocurrent with the light propagation direction. For the TE mode, the photocurrent is dominated by the LPGE contribution, which is independent of the light propagation direction. Figure 4a also shows that the CPGE and the LPGE photocurrents likely have different cut-off frequencies, which is probable because they originate from the surface and the bulk of the $Bi_2Se_3$, respectively. However, in our device their cut-off frequencies are too close to be reliably extracted and hence a conclusive analysis cannot be performed, but is possible with an optimized device design. The LPGE effect originates from the bulk of the TI and, as shown in Fig. 2, is highly anisotropic with respect to the crystalline axes. Therefore, its contribution can vary over a large range depending on the orientation of the TI flake relative to the waveguide, which is uncontrolled in our fabrication process. Figure 4b summarizes the measured results, for the TM mode and at a fixed frequency of 110 Hz, from four devices (A–D) with varying length $l$ of the TI between the longitudinal electrode pairs (Supplementary Fig. 4). The results indeed show a large variation in terms of the relative contribution of photocurrent between the CPGE and LPGE effects. In samples where LPGE overwhelms the CPGE, the total photocurrent will not reverse sign with the change of light propagation direction, but its magnitude will vary because of the reversal of the CPGE contribution. Characterization methods such as micro-Raman spectroscopy can be employed to determine the crystalline orientation of the $Bi_2Se_3$ flake. Therefore, it is possible to precisely align the orientation of the flake with the waveguide to minimize or even completely remove the LPGE contribution from the bulk (Fig. 2g) to obtain a highly asymmetric, helicity-dependent photoresponse with respect to the light propagation direction in the waveguide.

Our results demonstrate a helicity-dependent interface between the optical modes in a photonic waveguide and the surface electrons of topological insulators that directly converts the spin angular momentum of the guided photons to the spin-polarized photocurrent flowing on the surface of the TI. The helicity dependence of the interaction stems from the spin-momentum locking effects for both photons and electrons and their coupling in an integrated platform. More broadly, our device represents a way of implementing coupled spin–orbit interaction of both electrons and photons. The device can be utilized as an optically pumped source of spin-polarized current that may find application in spintronics. To this end, an important next step is to verify the spin polarization in the CPGE photocurrent with techniques such as magneto-optic Kerr effect[25,26] and non-local potentiometric measurement using magnetic contacts[27,28]. Furthermore, the device converts the path information of photon propagation to the spin polarization of the electrical current and thus can be used as an interface between photon qubits and spin qubits in a hybrid system of quantum information processing[5,29–33].

## Methods

**Device fabrication**. The $Bi_2Se_3$ flakes were exfoliated from a bulk crystal (obtained from HQ Graphene, Netherland) and transferred onto a silicon substrate with 450 nm thick $SiO_2$ layer. To explore how the photoresponse depends on sample crystalline direction, the flakes were first patterned into a circular shape using electron beam lithography and low power argon ion milling. A Ti(10 nm)/Al (400 nm)/Ti(10 nm)/Au(80 nm) metal multilayer was deposited as the top contacts using electron beam evaporator after an interface cleaning step using argon ion milling for 10 s.

The waveguide-integrated $Bi_2Se_3$ devices were fabricated on a silicon-on-insulator wafer with a 220 nm top silicon layer and a 3 µm buried oxide layer. The

underlying photonics layer was patterned using standard ebeam lithography and reactive ion etching processes. Subsequently, a 150 nm layer of hydrogen silsesquioxane was spin-coated and exposed with ebeam lithography to planarize the photonics layer. The bottom metal contacts were defined using ebeam lithography and Ti(5 nm)/Au(50 nm) was deposited using ebeam evaporator. Finally, exfoliated $Bi_2Se_3$ flakes were transferred onto the bottom contacts using a dry transfer method.

**Measurement**. All of the measurements were performed at room temperature in the atmosphere. The photoresponse was measured by modulating a CW laser source (Agilent 81940A) with a high-speed electro-optic modulator (Lucent 2623NA) driven by a lock-in amplifier (Stanford Research Systems, SR865A). To measure the polarization and crystalline direction dependence in the free-space configuration, an additional erbium-doped fiber amplifier was used after the modulator to amplify the optical power and the laser intensity was modulated at 10 kHz. When measuring the waveguide-integrated device, a fiber array was used to couple light into the device and collect the transmitted optical signal for alignment and calibration. Integrated grating couplers were fabricated to couple the TE and TM modes into the photonic device. A fiber polarization controller was used to control the optical polarization. The coupling efficiencies for the TE and the TM mode couplers are ~8 and ~6%, respectively. During the measurement, the output laser power was fixed at 4 mW and hence the devices' photoresponse is in the linear regime.

**Data availability**. The data that support the findings of this study are available from the corresponding author upon reasonable request.

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

## Acknowledgements

This work was supported by C-SPIN, one of the six centers of STARnet, a Semiconductor Research Corporation program, sponsored by MARCO and DARPA, and the National Science Foundation (Award No. ECCS-1351002). L.H. acknowledges the support of Doctoral Dissertation Fellowship provided by the Graduate School of the University of Minnesota. S.L. acknowledges the scholarship provided by the China Scholarship Council. Parts of this work were carried out in the University of Minnesota Nanofabrication Center which receives partial support from NSF through NNCI program, and the Characterization Facility which is a member of the NSF-funded Materials Research Facilities Network via the MRSEC program.

## Author contributions

L.H. and M.L. conceived the research. S.L. and L.H. fabricated the devices, performed the measurements and analyzed the results. L.H. and M.L. co-wrote the manuscript.

## Additional information

**Competing interests:** The authors declare no competing financial interests.

