## [Peer Review File · Nature Communications]

Editorial Note: This manuscript has been previously reviewed at another journal that is not operating a transparent peer review scheme. This document only contains reviewer comments and rebuttal letters for versions considered at Nature Communications. Mentions of prior referee reports have been redacted.

Reviewers' comments:

Reviewer #1 (Remarks to the Author):

The manuscript combines two interesting topics of chiral light-matter coupling and topological insulators. This is an interesting and novel achievement. The fundamental operational principle is laid out in a clear and precise manner, and the experimental demonstration is convincing and seems to be well conducted. The work is a nice demonstration of how solid-state physics and nanophotonics concepts are brought together leading to new physics. I recommend publication of the manuscript in Nature Comm. after the authors consider the following remarks:

- * the authors talk about the photon propagation as being chiral throughout the manuscript (e.g. in the abstract) and also refer to the waveguide as being chiral. This is a misuse of the terminology. In this case a circularly polarized beam of light reflecting off a mirror would also be chiral since reciprocity of Maxwell's equations implies that the forwardly propagating beam is the complex conjugate of the backwardly propagating beam. In the present experiment it is the light-matter interaction that is chiral, which is clearly explained in the cited literature. This distinction should be clarified in the manuscript.
- * The second sentence after the abstract: "The total field is spinning ..." was not very clear.
- * It was not entirely clear to me why the CPGE current is longitudinal (e.g. Fig. 3) rather than transverse. This point could be clarified. Also the sketch of Fig. 1e seems to indicate that the opposite is expected.
- * There are units missing on the x-axes in Figs. 3
- * There are a few minor wording/spelling misprints throughout the manuscript

Reviewer #2 (Remarks to the Author):

In the manuscript "Chiral interaction between spin-momentum locked photons and surface electrons in topological insulators" the authors have experimentally shown the circular photogalvanic effect (CPGE) induced by spin-momentum locked transverse spin of a waveguide mode. This result is quite interesting and is the first demonstration of spin-momentum locking transfer from evanescent waves to electrons in a topological insulator. I anticipate the result to be of importance for a broad community and is suitable for Nature Communications.

The authors have first measured dependence of photocurrent to the angle of incidence of light with respect to the Bi₂Se₃ crystal at $\lambda=1.55\mu\text{m}$. To this end, a device with 9 pairs of probes at different angles on a circular Bi₂Se₃ flake is fabricated. Photocurrent is measured at different incidence angles by rotating the crystal with respect to the incident beam and using the corresponding pair of probes. The dependence of different terms contributing to the total photocurrent, including CPGE and linear photogalvanic effect (LPGE), is extracted from these measurements. The authors determine which term arises from bulk effects and which from surface effects which agrees with the existing literature.

They continue with measuring evanescent field induced CPGE which is the main result of the paper. Devices are fabricated where a flake of Bi₂Se₃ is placed above a waveguide to interact with the evanescent field of its modes. Two pairs of probes are designed to measure the induced

photocurrent in longitudinal and transverse directions with respect to the waveguide. Photocurrent induced in the probes is measured for different directions of propagation for each of TE and TM modes of the waveguide. By comparing the results of different measurements they have shown the direction of induced CPGE is only dependent on direction of propagation of the waveguide mode, agreeing with spin-momentum locking of light in evanescent waves. By fabricating different devices, they have shown for certain devices the CPGE effect can be dominant, larger than LPGE by a factor of 3. But, for others it can be smaller than LPGE. They have attributed this to the randomness of the angle between the axes of the crystal and the waveguide as a result of their fabrication process, but, have not proved it. They also show that the induced CPGE is negligible in TE modes compared to TM modes.

The results are interesting both for showing the dependence of the photocurrent on crystal angle and showing the spin-momentum locked interaction. However, the following comments need to be addressed in their manuscript:

- Describing the guided modes with transverse spin as "chiral" modes is not correct. A chiral object cannot be mapped to its enantiomer by any rotation. However, in their experiment they show changing the direction of propagation of the TM mode excites the opposite spin of the surface states. Changing direction of propagation is equivalent to a 180-degree rotation. If the interaction was chiral the same spin of surface states should have been excited. The difference between a chiral dipole and spin-polarized dipole was first explained in *Optica* 3 (2), 118-126 2016.
- They have defined spin as $S \sim E \times E$. For a broad audience it would be good for the authors to mention that this is just the electric Stokes parameter evaluated for evanescent waves. Another important point is that this is only electric spin and magnetic spin has been omitted. The total spin is $S = -(E \times E - H \times H)$. Both TE and TM mode have transverse spin. However, in the studied structure the electric spin of TM mode interacts with electrons more effectively than the magnetic spin of TE mode.
- The error bars in the experiments are not discussed. Moreover, in Fig.2 d-g the fitted parameters are not presented and their agreement with the authors' theory and conclusions are not discussed. The authors should quantify the asymmetry and present figures of merit for the coupling. Is this more or less efficient than free space light induced spin-momentum locking in the topological insulator?
- Photon drag effect is not mentioned in the discussion of photocurrent sources. It doesn't seem to be a significant effect based on the presented results. However, because it would have the same effect as CPGE it's good to be mentioned.
- It is not discussed why the transverse photocurrent is different in amplitude for TE and TM mode in Fig.3 g.

The Authors' Response to Reviewers' Comments

Reviewer 1's comments:

The manuscript combines two interesting topics of chiral light-matter coupling and topological insulators. This is an interesting and novel achievement. The fundamental operational principle is laid out in a clear and precise manner, and the experimental demonstration is convincing and seems to be well conducted. The work is a nice demonstration of how solid-state physics and nanophotonics concepts are brought together leading to new physics. I recommend publication of the manuscript in Nature Comm. after the authors consider the following remarks.

Our response: We thank Reviewer 1 very much for the favorable comments and the critical and constructive advice. In the following, we address reviewer's comments point by point.

1. The authors talk about the photon propagation as being chiral throughout the manuscript (e.g. in the abstract) and also refer to the waveguide as being chiral. This is a misuse of the terminology. In this case a circularly polarized beam of light reflecting off a mirror would also be chiral since reciprocity of Maxwell's equations implies that the forwardly propagating beam is the complex conjugate of the backwardly propagating beam. In the present experiment it is the light-matter interaction that is chiral, which is clearly explained in the cited literature. This distinction should be clarified in the manuscript.

Our response: We thank the reviewer to point out that claiming the waveguide mode itself as being "chiral" may be controversial. In fact, the circularly polarized evanescent fields have been described as "chiral" in many other literatures (for example, Coles, R., Price, D., Dixon, J., Royall, B., Clarke, E., Kok, P., Skolnick, M., Fox, A. & Makhonin, M. Chirality of nanophotonic waveguide with embedded quantum emitter for unidirectional spin transfer. *Nat. Commun.* 7 (2016)). However, as also pointed by the 2nd reviewer, the locked optical spin angular momentum and linear momentum in counter-propagating directions are not in mirror symmetry and can be mapped to each other by π rotation. Therefore, the evanescent field of the mode is only "helical" but not "chiral". (Please also see our response to the 2nd reviewer's first comment.)

To be consistent with the correct definition of "chiral", we have dropped the use of "chiral waveguide mode" in the revised manuscript accordingly. Instead, we emphasize the spin-momentum locking of the waveguide mode and the interaction with TI surface states.

2. The second sentence after the abstract: "The total field is spinning ..." was not very clear.

Our response: To clarify, we have revised this sentence and the one before it to:

"The electric field of a non-paraxial optical beam or laterally confined optical modes in fibers or waveguides is no longer purely transverse but has a longitudinal field component, which has a $\pm\pi/2$ phase shift relative to the transverse field component. Therefore, the total electric field is spinning along an axis transverse to the light propagation direction and elliptically polarized in the propagation plane^{1,2}."

3. It was not entirely clear to me why the CPGE current is longitudinal (e.g. Fig. 3) rather than transverse. This point could be clarified. Also the sketch of Fig. 1e seems to indicate that the opposite is expected.

Our response: The CPGE current is longitudinal because of the spin-momentum locking of both the waveguide mode and the TI surface states. The photons in TM mode carry transverse

SAM (along the x -axis in Fig. 3b), which selectively excite surface electrons with spin aligned along the same direction (x -axis). Due to the spin-momentum locking effect of surface electrons, the induced CPGE current is perpendicular to electron spin orientation (x -axis), which is in the longitudinal direction (z -axis). We have further clarified this point in the revised manuscript as follows:

“Because the photons in the TM mode are elliptically polarized and carry transverse electric SAM (along x -axis), through the CPGE effect they will selectively excite surface electrons with their spin aligned with the optical SAM and induce a photocurrent flowing in the longitudinal direction (z -axis) due to the spin-momentum locking effect of topological surface states.”

Fig. 1e means to show the device has an optical input and an electrical output. We agree with the reviewer that it can be misleading. We have revised it to show that the output photocurrent flows along the same direction as the light propagation.

4. There are units missing on the x-axes in Figs. 3.

Our response: We have added the missing x -axes label “Frequency (Hz)” in Fig. 3d-g.

5. There are a few minor wording/spelling misprints throughout the manuscript

Our response: We appreciate the reviewer to point out this. We have proofread and corrected these typos in our manuscript.

Reviewer 2's comments:

In the manuscript “Chiral interaction between spin-momentum locked photons and surface electrons in topological insulators” the authors have experimentally shown the circular photogalvanic effect (CPGE) induced by spin-momentum locked transverse spin of a waveguide mode. This result is quite interesting and is the first demonstration of spin-momentum locking transfer from evanescent waves to electrons in a topological insulator. I anticipate the result to be of importance for a broad community and is suitable for Nature Communications.

The authors have first measured dependence of photocurrent to the angle of incidence of light with respect to the Bi₂Se₃ crystal at $\lambda=1.55\mu\text{m}$. To this end, a device with 9 pairs of probes at different angles on a circular Bi₂Se₃ flake is fabricated. Photocurrent is measured at different incidence angles by rotating the crystal with respect to the incident beam and using the corresponding pair of probes. The dependence of different terms contributing to the total photocurrent, including CPGE and linear photogalvanic effect (LPGE), is extracted from these measurements. The authors determine which term arises from bulk effects and which from surface effects which agrees with the existing literature.

They continue with measuring evanescent field induced CPGE which is the main result of the paper. Devices are fabricated where a flake of Bi₂Se₃ is placed above a waveguide to interact with the evanescent field of its modes. Two pairs of probes are designed to measure the induced photocurrent in longitudinal and transverse directions with respect to the waveguide. Photocurrent induced in the probes is measured for different directions of propagation for each of TE and TM modes of the waveguide. By comparing the results of different measurements they have shown the direction of induced CPGE is only dependent on direction of propagation of the waveguide mode, agreeing with spin-momentum locking of light in evanescent waves. By fabricating different devices, they have shown for certain devices the CPGE effect can be dominant, larger than LPGE by a factor of 3. But, for others it can be smaller than LPGE. They have attributed this to the randomness of the angle between the axes of the crystal and the waveguide as a result of their fabrication process, but, have not proved it. They also show that the induced CPGE is negligible in TE modes compared to TM modes.

The results are interesting both for showing the dependence of the photocurrent on crystal angle and showing the spin-momentum locked interaction. However, the following comments needs to be addressed in their manuscript.

Our response: We thank Reviewer 2 very much for the positive comments and the critical and constructive advice. In the following, we address the reviewer's comments point by point.

1. Describing the guided modes with transverse spin as “chiral” modes is not correct. A chiral object cannot be mapped to its enantiomer by any rotation. However, in their experiment they show changing the direction of propagation of the TM mode excites the opposite spin of the surface states. Changing direction of propagation is equivalent to a 180-degree rotation. If the interaction was chiral the same spin of surface states should have been excited. The difference between a chiral dipole and spin-polarized dipole was first explained in *Optica* 3 (2), 118-126 2016.

Our response: We thank the reviewer for pointing out that claiming a waveguide mode itself as being “chiral” may be inappropriate. After careful consideration, we agree and have revised the manuscript accordingly. We also agree with the reviewer that the use of “chirality” is

controversial in our case and in related situations of light-matter interactions that have been described as “chiral” in literatures as listed below.

1. Lodahl, P., Mahmoodian, S., Stobbe, S., Rauschenbeutel, A., Schneeweiss, P., Volz, J., Pichler, H. & Zoller, P. Chiral quantum optics. *Nature* **541**, 473-480 (2017).
2. Söllner, I. *et al.* Deterministic photon-emitter coupling in chiral photonic circuits, *Nat. Nanotechnol.* **10**, 775-778 (2015).
3. Petersen, J., Volz, J. & Rauschenbeutel, A. Chiral nanophotonic waveguide interface based on spin-orbit interaction of light. *Science* **346**, 67–71 (2014).
4. Coles, R., Price, D., Dixon, J., Royall, B., Clarke, E., Kok, P., Skolnick, M., Fox, A. & Makhonin, M. Chirality of nanophotonic waveguide with embedded quantum emitter for unidirectional spin transfer. *Nat. Commun.* **7** (2016).

To be consistent with the correct definition of chirality, we have dropped the use of “chiral” throughout the manuscript and instead emphasize the spin-momentum locking aspect.

2. They have defined spin as $\mathbf{S} \sim \mathbf{E}^* \times \mathbf{E}$. For a broad audience it would be good for the authors to mention that this is just the electric Stokes parameter evaluated for evanescent waves. Another important point is that this is only electric spin and magnetic spin has been omitted. The total spin is $\mathbf{S} = -[\mathbf{E}^* \times \mathbf{E} + \mathbf{H}^* \times \mathbf{H}]$. Both TE and TM mode have transverse spin. However, in the studied structure the electric spin of TM mode interacts with electrons more effectively than the magnetic spin of TE mode.

Our response: We completely agree with reviewer that the spin defined in the manuscript $\mathbf{S} = -[i\epsilon_0/2\omega]\mathbf{E}^* \times \mathbf{E}$ refers to electric spin. It is the electric spin that gives rise to the leading term of the electromagnetic perturbation Hamiltonian ($H_{em} \sim -\frac{e}{mc}\mathbf{p} \cdot \mathbf{A}$), which governs the optical transitions in the material. We have clarified the definition in the manuscript accordingly.

3. The error bars in the experiments are not discussed. Moreover, in Fig.2 d-g the fitted parameters are not presented and their agreement with the authors’ theory and conclusions are not discussed. The authors should quantify the asymmetry and present figures of merit for the coupling. Is this more or less efficient than free space light induced spin-momentum locking in the topological insulator?

Our response: The original data presented in Fig. 2c and other figures are measured from a single device with very low measurement uncertainty and variation, so there is no statistical error to be presented. In Fig. 2d-g, the parameters are obtained by fitting the original data in Fig. 2c with very small uncertainties. For example, in the figure below we plot out the four coefficients (C, L_1, L_2, D) with error bars, which represent the confidence bounds of 95%. The error bars are smaller than the symbol size. The small fitting uncertainties can also be seen from Fig. 2c, where the measured data match extremely well with the fitted curves using equation (1) in the manuscript. In the revised manuscript, we have mentioned the small fitting uncertainties.

In Fig. 2. f-g, we fit the results with the formula $j_y = a * \sin[(2\pi\phi/120) + \varphi_0] + b$, where the coefficient $2\pi\phi/120$ characterizes the periodicity of photocurrent with the crystalline orientation, which has three-fold rotation symmetry. φ_0 accounts for the offset angle between the sample crystalline and the first pair of metal contacts used as the 0-degree reference.

The coefficient a is photocurrent amplitude and b accounts for a small polarization-independent dc background due to thermal effect. The fitted value for a is 9.91 and 56.94 $\text{pA}\cdot\text{W}^{-1}\cdot\text{cm}^2$ for L_2 and D , respectively, whereas the fitted b is 2.12 and 0.57 $\text{pA}\cdot\text{W}^{-1}\cdot\text{cm}^2$ for L_2 and D , respectively. The result suggests that the corresponding photocurrent can flip polarity completely if rotating the crystalline orientation. The fitted values φ_0 for L_2 and D (91.2° and 90.8° , respectively) are very close. This agreement shows that the two effects indeed have the same crystalline dependence and share a common origin. Furthermore, the three-fold rotation symmetry observed in L_2 and D may also suggest that the corresponding microscopic mechanism does not include the surface bands, which only exist near the Γ point and are isotropic.

A fair comparison in coupling efficiencies between the free-space device and waveguide-integrated device is challenging, as the two devices have distinct geometries and the effective device areas can be poorly estimated. Nevertheless, we roughly estimate the efficiencies in CPGE photocurrent generation for the two devices by normalizing the measured CPGE current to optical power and effective flake area. The estimated efficiencies are $\sim 0.05 \text{ mA}\cdot\text{W}^{-1}\cdot\mu\text{m}^{-2}$

and $\sim 0.22 \text{ mA}\cdot\text{W}^{-1}\cdot\mu\text{m}^{-2}$ for the free-space and the waveguide device, respectively, which suggests the coupling through evanescent fields may be more efficient. In fact, this conclusion should be correct since the free-space configuration suffers from strong reflection from the material surface, which lowers the total photoresponse. More systematic study is required in order to make more conclusive comparison.

4. Photon drag effect is not mentioned in the discussion of photocurrent sources. It doesn't seem to be a significant effect based on the presented results. However, because it would have the same effect as CPGE it's good to be mentioned.

Our response: We agree with the reviewer that photon drag effect can in principle generate a helicity-dependent photocurrent through circular photon drag effect (CPDE), which involves the transfer of both optical angular momentum and linear momentum. However, in contrast to the CPGE whose magnitude is proportional to $\sin \theta$ (where θ is light incident angle), the transverse photocurrent induced by CPDE is proportional to $\sin 2\theta$. Therefore, the contribution from CPDE to the photocurrent can be excluded by measuring the helicity-dependent photocurrent as a function of incidence angle, which has recently been experimentally demonstrated in a similar material system $(\text{Bi}_{1-x}\text{Sb}_x)\text{Te}_3$ by Samarth and Awschalom groups and reported in Pan *et al.* We have added more discussions and the reference as listed below on the photon drag effect in the revised manuscript.

1. Pan, Y., Wang, Q., Yeats, A., Pillsbury, T., Flanagan, T., Richardella, A, Zhang, H., Awschalom, D., Liu, C., Samarth, N., Helicity dependent photocurrent in electrically gated $(\text{Bi}_{1-x}\text{Sb}_x)\text{Te}_3$ thin films. Preprint at <https://arxiv.org/abs/1706.04296> (2017).

5. It is not discussed why the transverse photocurrent is different in amplitude for TE and TM mode in Fig.3 g.

Our response: As analyzed in the manuscript, the transverse photocurrent is attributed to the LPGE effect. The difference in transverse photocurrent amplitude for TE and TM mode may be attributed to the distinctive vertical optical field confinement for the two modes. Specifically, the TM mode is weakly confined in vertical direction compared to TE mode, as can be seen from the simulated mode profiles below (note that the electric field is normalized such that the two modes carry the same optical power and the color scale is the same in the two plots). Therefore, under TM mode excitation, stronger electric field on the Bi_2Se_3 bottom surface is expected and leads to a larger transverse photocurrent in the experiment. However, a more quantitative analysis is difficult and requires more comprehensive study of the LPGE, as the transverse photocurrent due to LPGE also depends on light polarization and the crystalline orientation (e.g. Junck, A., Theory of photocurrents in topological insulators. Ph.D. thesis, im Fachbereich Physik der Freien Universität Berlin eingereichte (2015)).

In the revised manuscript, we have added a brief comment on the difference in transverse photocurrent responses for the two modes.

REVIEWERS' COMMENTS:

Reviewer #1 (Remarks to the Author):

I think the authors have adapted well to the comments of the reviewers and I support publication. Only one unclarity remains: the authors have corrected the statements that light propagation in the waveguide is chiral. Indeed this point was pointed out by both referees.

However the interaction between light and matter can be considered chiral, since the forward propagating light and the backward propagating light (the mirror image) interact differently. That is the operational definition of chiral light-matter interaction and chiral quantum optics (see, e.g., Ref. 5 + 7), which is taking place in the current experiment in the interaction between light and electrons. I would encourage the authors to clarify this distinction between chiral light transport and chiral light-matter interaction.

Reviewer #2 (Remarks to the Author):

The comments have been addressed and the manuscript can be published.

Reviewer 1's comments:

I think the authors have adapted well to the comments of the reviewers and I support publication.

Only one unclarity remains: the authors have corrected the statements that light propagation in the waveguide is chiral. Indeed this point was pointed out by both referees.

However the interaction between light and matter can be considered chiral, since the forward propagating light and the backward propagating light (the mirror image) interact differently. That is the operational definition of chiral light-matter interaction and chiral quantum optics (see, e.g., Ref. 5 + 7), which is taking place in the current experiment in the interaction between light and electrons. I would encourage the authors to clarify this distinction between chiral light transport and chiral light-matter interaction.

Our response: We agree with reviewer 1 that the forward/backward propagating mode in our system interacts differently with surface electrons in TI, thus the coupling indeed falls into the operational definition of chiral light-matter interaction. In fact, this definition of chiral light-matter interaction has been adopted in many recent literatures, some of which are cited in the references. However, one may make the opposite conclusion if follows the canonical definition that a chiral object cannot be mapped to its mirror image through rotation and translation alone. For example, note that in our case the optical spin is pointing in the waveguide plane (x -axis), and when both electron and photon spins are considered, the device under forward light input situation can be transformed to the situation of backward light input with an in-plane π rotation alone, as illustrated in panel **a**, such a device is achiral. (Similarly, the system in Ref. 7 should also be achiral.) This is in contrast to the situation in Ref. 5, where the transverse optical spin is perpendicular to the waveguide plane (y -axis). Consequently, when only photon spin is considered, the backward propagation cannot be mapped to the forward propagation with an in-plane π rotation but a mirror reflection, as shown in panel **b**. This should be considered as chiral.

In fact, whether or not the interaction is chiral may depends on the definition adopted and thus controversial (as we have received conflicting comments from the two reviewers). However, this argument does not affect the main conclusion of our work. Therefore, to avoid any further arguments on the terminology, we would like to avoid claiming the waveguide mode-TI coupling as being chiral.

Figure 1 a. Situation of our waveguide-TI devices, where the photon spin is in the x-z plane. Red symbols mark the photon k and spin, purple symbols mark the electrical current and spin. The backward light propagation situation can be transformed to the forward propagation direction by an in-plane π rotation, reversing both the photon and the electron spins. In contrast, the mirror reflection does not reverse the electron spin. Therefore, the interaction is achiral. **R b.** Situation of out-of-plane photon spin as in reference 5. In this situation, the forward and backward propagation situation are mirror reflection of each other. Therefore, the system is chiral.